# Validation of Indirect Calorimetry in Children Undergoing Single-Limb Non-Invasive Ventilation: A Proof of Concept, Cross-Over Study

**DOI:** 10.3390/nu16020230

**Published:** 2024-01-11

**Authors:** Veronica D’Oria, Giulia Carla Immacolata Spolidoro, Carlo Virginio Agostoni, Cinzia Montani, Ludovica Ughi, Cristina Villa, Tiziana Marchesi, Giovanni Babini, Stefano Scalia Catenacci, Giada Donà, Marta Guerrini, Giovanna Chidini, Edoardo Calderini, Thomas Langer

**Affiliations:** 1Pediatric Intensive Care Unit, Fondazione IRCCS Ca’ Granda Ospedale Maggiore Policlinico, 20122 Milan, Italy; veronica.doria@policlinico.mi.it (V.D.); cinzia.montani@policlinico.mi.it (C.M.); ludovica.ughi@policlinico.mi.it (L.U.); tiziana.marchesi@policlinico.mi.it (T.M.); stefano.scaliacatenacci@policlinico.mi.it (S.S.C.); giada.dona@policlinico.mi.it (G.D.); giovanna.chidini@policlinico.mi.it (G.C.); edoardocalderini@gmail.com (E.C.); thomas.langer@unimib.it (T.L.); 2Department of Clinical Sciences and Community Health, University of Milan, 20122 Milan, Italy; giulia.spolidoro@unimi.it; 3Pediatric Unit, Fondazione IRCCS Ca’ Granda Ospedale Maggiore Policlinico, 20122 Milan, Italy; 4Department of Anesthesiology, Intensive Care and Emergency, Fondazione IRCCS Ca’ Granda Ospedale Maggiore Policlinico, 20122 Milan, Italy; cristina.villa@policlinico.mi.it (C.V.); giovanni.babini@policlinico.mi.it (G.B.); 5Department of Healthcare Professions, Fondazione IRCCS Ca’ Granda Ospedale Maggiore Policlinico, 20122 Milan, Italy; marta.guerrini@policlinico.mi.it; 6School of Medicine and Surgery, University of Milan-Bicocca, 20900 Monza, Italy

**Keywords:** indirect calorimetry, resting energy expenditure, non-invasive ventilation, critically ill children, pediatric intensive care

## Abstract

Background. The accurate assessment of resting energy expenditure (REE) is essential for personalized nutrition, particularly in critically ill children. Indirect calorimetry (IC) is the gold standard for measuring REE. This methodology is based on the measurement of oxygen consumption (VO_2_) and carbon dioxide production (VCO_2_). These parameters are integrated into the Weir equation to calculate REE. Additionally, IC facilitates the determination of the respiratory quotient (RQ), offering valuable insights into a patient’s carbohydrate and lipid consumption. IC validation is limited to spontaneously breathing and mechanically ventilated patients, but it is not validated in patients undergoing non-invasive ventilation (NIV). This study investigates the application of IC during NIV-CPAP (continuous positive airway pressure) and NIV-PS (pressure support). Methods. This study was conducted in the Pediatric Intensive Care Unit of IRCCS Ca’ Granda, Ospedale Maggiore Policlinico, Milan, between 2019 and 2021. Children < 6 years weaning from NIV were enrolled. IC was performed during spontaneous breathing (SB), NIV-CPAP, and NIV-PS in each patient. A Bland–Altman analysis was employed to compare REE, VO_2_, VCO_2_, and RQ measured by IC. Results. Fourteen patients (median age 7 (4; 18) months, median weight 7.7 (5.5; 9.7) kg) were enrolled. The REE, VO_2_, VCO_2_, and RQ did not differ significantly between the groups. The Limits of Agreement (LoA) and bias of REE indicated good agreement between SB and NIV-CPAP (LoA +28.2, −19.4 kcal/kg/day; bias +4.4 kcal/kg/day), and between SB and NIV-PS (LoA −22.2, +23.1 kcal/kg/day; bias 0.4 kcal/kg/day). Conclusions. These preliminary findings support the accuracy of IC in children undergoing NIV. Further validation in a larger cohort is warranted.

## 1. Introduction

The nutritional status in critically ill pediatric patients is considered a basic prognostic factor in terms of mortality, morbidity, complications, and outcomes [1]. Metabolic alterations are common and difficult to prevent in patients admitted to the pediatric intensive care unit (PICU) and vary from hypometabolism (Measured Resting Energy Expenditure [MREE] < 90% predicted) to hypermetabolism (MREE > 110% predicted) [2]. Recent insights have underscored the adverse impact of malnutrition, encompassing both underfeeding and overfeeding, on patient outcomes. The optimization of nutritional support assumes a pivotal role in clinical practice, thereby emphasizing the importance of assessing resting energy expenditure (REE) in PICU-admitted patients [3,4,5,6,7,8,9,10].

Critically ill children have an increased risk of nutritional therapy failure because of the changes in energy consumption, as well as in energy needs, which may both vary over time throughout the course of a child’s hospitalization [4,5].

A recent study showed that undernutrition at PICU admission was predictive of 60-day mortality and a longer time to discharge alive from the PICU [6]. Moreover, a recent meta-analysis confirmed the importance of avoiding both undernutrition and overnutrition in this population, since both conditions are linked to a higher mortality incidence, longer PICU stay, and longer mechanical ventilation usage and duration [7].

The basal metabolic rate (BMR) represents the energy required to sustain vital functions. It is measured in a recumbent position, in a thermo-neutral environment, fasting for 12–18 h, when the subject has awakened before starting daily activities. In practice, this is impossible to measure during infancy and most of childhood. According to ESPGHAN/ESPEN/ESPR guidelines, REE does not differ more than 10% from BMR; therefore, REE is usually measured instead of BMR in this population [8]. REE can be estimated using various formulas; nevertheless, several studies have demonstrated that commonly used predictive equations often provide inaccurate estimates, leading to the under- or overestimation of energy requirements [9,10,11]. As a result, the measurement of REE through indirect calorimetry (IC) has emerged as the gold standard method. IC’s precise measurement of energy expenditure permits the delivery of personalized nutritional support throughout the disease trajectory, enabling daily guided nutritional prescriptions [12,13,14]. Consequently, IC has gained a pivotal and well-established role in the PICU setting [15]. This methodology is based on the measurement of inspired and expired oxygen and carbon dioxide concentrations [16]. Oxygen consumption (VO_2_ L/min) and carbon dioxide production (VCO_2_ L/min) are then integrated into the Weir equation to calculate energy expenditure [3]. Additionally, IC facilitates the determination of the respiratory quotient (RQ), offering valuable insights into a patient’s carbohydrate and lipid consumption [17].

Currently, indirect calorimetry (IC) can only be conducted in patients who are either spontaneously breathing or mechanically ventilated. For spontaneously breathing patients, IC entails the use of a transparent “canopy” helmet positioned over the subject’s head while the subject remains in a supine position. The canopy incorporates two openings, one at the top and another at the bottom of the helmet. The lower opening is connected to the calorimeter via a tubing system. Furthermore, the calorimeter is equipped with a flow generator that enables the adjustment of the aspiration flow within the helmet. This adjustment ensures that the ambient air is mixed with the subject’s exhalation. In patients undergoing invasive ventilation, sampling is accomplished by connecting the calorimeter directly to the expiratory circuit of the ventilator [3].

Non-invasive ventilation (NIV) is commonly employed in patients requiring chronic respiratory support, and those experiencing acute respiratory failure [18]. In recent years, NIV through single-limb circuits with intentional leaks have gained popularity in the pediatric population [19].

Unfortunately, a validated method for the measuring of REE via IC in non-invasively ventilated patients is not yet available and only limited data exist on the use of IC in adult subjects undergoing NIV [20,21].

Therefore, NIV currently prevents the delivery of personalized nutritional support guided by IC in children admitted to the PICU.

The aim of our study was to validate the measurement of IC in pediatric patients undergoing NIV through a single-limb circuit with intentional leaks.

Our hypothesis was that REE measurement by IC would be reliable in children undergoing NIV.

## 2. Materials and Methods

This study is an interventional, cross-over, single center study conducted in the Pediatric Intensive Care Unit (PICU) of IRCCS Ca’ Granda, Ospedale Maggiore Policlinico of Milan, between January 2019 and June 2021. The research was carried out in accordance with the principles stated in the Declaration of Helsinki for Research on Human Subjects. This study was approved by the local ethics committee (Approval Number date 10 January 2019) and it is registered on ClinicalTrials.gov (ID NCT03824249).

### 2.1. Patients’ Enrollment

Enrolment in this study was conducted through a selective invitation process. We included pediatric patients aged 1 month to 5 years admitted to the PICU who were undergoing the weaning phase from NIV, i.e., patients that could tolerate phases of spontaneous breathing, were used to NIV, and were no longer on oxygen supplementation. Additionally, enrolment required signed informed consent from the parents of the eligible children. Exclusion criteria were unreliability of IC measurement due to lack of a standard condition, such as average RQ < 0.67, a non-linear trend of the exam, necessity to interrupt the study due to children’s experience of discomfort.

At study enrolment, the following information was collected: demographic data, medical history, vital signs (blood pressure, heart rate, peripheral oxygen saturation-SpO_2_, respiratory rate), NIV settings, nutritional data (type of nutrition and quantity, time of delivery), patient distress assessed using the COMFORT-B scale [22].

### 2.2. Measurements

Indirect calorimetry (IC) was performed to measure REE in enrolled patients. All measurements were performed with a VMAX Encore indirect calorimeter (Carefusion, Yorba Linda, CA, USA) using the canopy mode (Figure 1).

For every subject, measurements of REE were repeated in all the following conditions:Spontaneously breathing (gold standard).NIV-CPAP with a positive end expiratory pressure (PEEP) of 4 cmH_2_O: CPAP is applied via NIV, but no respiratory support is performed.NIV-PS with a pressure support (PS) of 8 cmH_2_O and a PEEP of 4 cmH_2_O: CPAP is applied via NIV with an additional inspiratory pressure support (PS).

NIV was carried out with a single-limb circuit with intentional leaks (Figure 2) using a turbine-driven ventilator (Astral 150 [ResMed, Vimercate, Italy] or Trilogy 150 [Philips Respironics, Milan, Italy] or Garbin [Linde, Arluno, Italy]), and a vented nasal mask (Respireo Soft Child-472711, [Air Liquide Medical System S.r.l, Brescia, Italy]), appropriate for patients’ size.

Prior to the performance of IC, total minute flow (minute ventilation, intentional leaks, non-intentional leaks) delivered by the ventilator was measured. This allowed us to know the gas volume delivered to the canopy helmet in one minute.

Total flow measurement was recorded by a pneumotachograph (Hans Rudolph Series 371, HANS RUDOLPH, Inc., Kansas City, MO, USA) placed across the ventilator exit and afterwards analyzed with dedicated software (ICU-Lab 3.1, KleisTEK Engineering, Bari, Italy). The obtained value enabled a personalized setting of IC aspiration flow, in order to optimize REE measurement. Considering previous studies [23], the aspiration flow was set at 125% of total flow measured by pneumotachograpy. IC was performed in the three experimental conditions (SB, NIV-CPAP, and NIV-PS) in randomized order. The randomization of the sequence was performed to avoid the risk of carry-over effect.

Each measurement lasted at least 20 min. Subjects undergoing IC were awake, but in resting condition and were lying in supine position. The three IC measurements were performed consecutively.

At the end of every IC measurement, the mean values of the following parameters were recorded: VO_2_ (mL/min), VCO_2_ (mL/min), RQ (VCO_2_/VO_2_), and REE (kcal/kg/day).

### 2.3. Randomization

The randomization of the test sequence was performed via “Research Randomizer” (Version 4.0) online software (https://www.randomizer.org/, accessed on 10 December 2018). No bias was expected in group assignment process, as patients’ REE were tested in each condition. All IC measurements were performed in consecutive order.

### 2.4. Statistical Analysis

Sample size was calculated considering the primary aim of the research using MedCalc 18.10.2 software (MedCalc Software bvba, Ostend, Belgium), based on the Bland–Altman method, which evaluates agreement between two measurement methods. According to Sirala et al. [20], the average difference in REE (kcal/day) measured in spontaneously breathing subjects (gold standard) and in NIV-supported subjects, was 6 kcal/day. Based on these data, a sample size of 20 was calculated considering a minimum difference in REE of 6 kcal, assuming a standard deviation of 0.50 with a desired power of 0.99 and an alpha error of 0.05.

### 2.5. Data Analysis 

Primary endpoint was to verify the agreement between REE (kcal/kg/day) measured in patients during spontaneous breathing (SB), the gold standard, and REE measured in patients during NIV-CPAP (continuous positive airway pressure). Secondary endpoint was to verify the agreement between REE (kcal/kg/day) measured in patients during SB and REE measured in patients undergoing NIV-PS (pressure support).

Continuous variables were described using mean and standard deviation (SD) or median and interquartile range (IQR), according to the normality of distribution. ANOVA test or Kruskal–Wallis test were applied to compare VO_2_, VCO_2_, and RQ (VCO_2_/VO_2_) between groups (SB, NIV-CPAP, and NIV-PS). Agreement was tested via Bland–Altman analysis: this method evaluates the agreement between two measurement methods, gold standard method vs. new method (IC performed in spontaneously breathing subjects vs. IC performed during NIV-CPAP/PS, in the case of our study).

Statistical analysis and the creation of graphs and representations were performed using GraphPad Prism 5.00.288.

## 3. Results

A total of 27 children aged 2–21 months, whose parents gave informed consent, were enrolled. Out of 27, 10 subjects dropped out from the study before performing IC. The specific reason for dropping out included the early discharge of the subjects from the PICU and the impossibility of performing the exam on the patient due to poor compliance/limited tolerance to the canopy helmet (e.g., children not in resting condition). The full study was completed by 17 subjects, of which 3 were excluded from the analysis after the evaluation of the IC exam, as described in the exclusion criteria. A total of 14 patients were included in our analysis, and specifically in 12 patients, IC was recorded in the SB, CPAP, and PS conditions. In two patients, IC was not recorded in the CPAP conditions, due to technical problem occurring during the measurement performed in the CPAP mode (Figure 3). The total number of subjects included was 14, of which 11 were males (78%).

The median age of the subjects analyzed was 7 (4; 18) months, the median weight was 8 (5; 10) kg, with a median length of 65 (58–79) cm. The reasons for PICU admission were bronchiolitis (6 subjects), pneumonia (4), post-surgery respiratory distress (3), laryngotracheitis (1). The median PICU length of stay (LOS) was 7 (3; 11) days. Vital signs (heart rate, peripheral oxygen saturation, and respiratory rate) were stable and within the respective normal ranges during the experiments (Table 1 and Table 2).

A comparison between the IC during spontaneous breathing and the two NIV conditions was analyzed. Since each patient underwent three consecutive measurements, a paired *t*-test was performed. To verify the possibility of measuring REE through IC during NIV and to validate these measurements, a Bland–Altman analysis was performed. Indirect calorimetry during the CPAP and PS conditions was compared to the gold standard method, i.e., IC during spontaneous breathing. The data are shown in Table 3 and Table 4, respectively.

Besides a slightly lower VCO_2_ measured in the CPAP mode, no other statistically significant differences were observed between the measurements performed during SB and CPAP or PS. All the RQ values were within the physiological range (0.67–1.1).

Figure 4 summarizes the Bland–Altman analysis performed for the REE. Overall, the agreement between the IC measurements performed during SB and during NIV-CPAP was acceptable. Similar results were found comparing the IC measurements during SB with IC performed during NIV-PS (Figure 4).

Considering the values of REE, the bias was lower (0.4 vs. 4.4 kcal/kg/day) and the Limits of Agreement were closer in the PS mode, therefore indicating a better agreement with IC performed during SB (gold standard method). Similar results were obtained regarding the comparison of gas exchange and RQ between SB and CPAP as well as between SB and PS (Figure 5, Figure 6 and Figure 7).

## 4. Discussion

Nutrition plays a pivotal role in critically ill children. Tailored nutrition, especially in terms of energy and protein, is an important prognostic factor and influences different outcomes such as the length of stay, morbidity, the risk of infection, and mortality [7]. Prescribing a personalized nutritional support starts from an accurate REE measurement, which currently requires IC. To date, IC is validated in spontaneously breathing and in mechanically ventilated patients, but not during NIV [1,3].

The aim of our study was to compare REE measured by IC in spontaneously breathing subjects (gold standard) with REE measured during NIV in a pediatric population weaning from NIV.

To the best of our knowledge, this is the first study critically appraising the possibility to perform IC in children undergoing NIV with a single-limb circuit. Other studies have investigated IC-guided REE assessment during NIV in adults with similar conclusions [20,21]. Compared to these studies, we have first measured the non-intentional leaks from the ventilator using the pneumotachograph and then set the IC aspiration flow at 125% of total ventilator flow, as suggested by an in vitro study published by Smallwood et al. [23]. This study investigated the agreement between VO_2_ and VCO_2_ measurements performed via IC and the reference values generated by an NIV simulator in room air. The authors demonstrated how the IC aspiration flow setting (V_IC_) needs to consider the ventilator total flow (V_VENT_) in order to obtain an accurate gas exchange measurement. Specifically, they suggested that a *sample factor* (V_IC_/V_VENT_) equal to 1.25 is linked to the best accuracy.

Our results show an acceptable agreement between the REE measured during spontaneous breathing (SB) and the REE measured during NIV-CPAP or NIV-PS. According to the Bland–Altman method, the bias between SB vs. NIV-PS was only 0.4 kcal/kg/day, while the bias between SB vs. NIV-CPAP was 4 kcal/kg/day. This is not surprising since the CPAP mode should not significantly alter the cost of breathing and, consequently, the REE measurement. Moreover, the upper and lower LoA were similar in SB vs. NIV-CPAP or NIV-PS (Table 3 and Table 4) and the points were well distributed around the bias line (Figure 4).

Overall, these results indicate a good accuracy of IC in pediatric subjects undergoing NIV. However, in some measurements, a marked difference was observed, suggesting that caution should be used when interpreting the data.

We observed a better agreement between the SB and NIV-PS mode compared to the SB and NIV-CPAP mode. Moreover, there was no significant difference between the calculated COMFORT-B score for the two ventilatory modes compared to SB (*p* > 0.05), even if a limit of this evaluation score in children should be considered.

This result could be explained by the observation that the subjective adaptation to the NIV-PS mode was better than the adaptation to NIV-CPAP. However, this is not of clinical relevance, since the REE measured during SB was comparable with the REE measured during NIV both in CPAP and in PS.

Our findings are in line with Sirala et al., who observed a good agreement between the REE measured in SB vs. the REE measured during NIV [20]. Particularly, the authors aimed at assessing IC in individuals supported by NIV with an intentional leak circuit, and they recruited 12 healthy adults. The resting energy expenditure was initially measured during SB and subsequently during NIV support. During NIV, the participants received support in the Pressure-Controlled Ventilation (PCV) mode, using a full-face mask. The Peak Inspiratory Pressure (PIP) and Positive End-Expiratory Pressure (PEEP) settings were customized based on the ability of each individual to adapt to the imposed pressure. Moreover, the IC aspiration flow was set to two different standardized flow rates: 40 L/min and 80 L/min. The findings of the study indicated a good level of accuracy for IC-guided REE measurement during NIV-PVC. Particularly, the REE varied from −2% to +3%, suggesting its potential applicability in healthy adults undergoing NIV. Importantly, a good agreement was found between the key variables (VO_2_, VCO_2_, and RQ) measured during SB and during NIV. Nevertheless, the clinical relevance of the reported results is limited by different factors. The authors employed two standardized IC flow rates, not considering the patients’ minute and volume ventilation. Moreover, the study was conducted on healthy adults, thus preventing the feasibility of the method on pediatric patients. This is especially true if we consider that, at this age, many characteristics vary: the minute ventilation is lower and intentional leaks are usually higher. For these reasons, REE may be underestimated.

Other studies have investigated the measurement of REE during NIV in subjects with chronic diseases. Georges et al. measured REE in 16 adults with amyotrophic lateral sclerosis (ALS), and they found that NIV could acutely decrease the REE in ALS patients with ALS-related chronic respiratory insufficiency [21]. Particularly, the authors compared the two ventilation modes and found a reduction in REE during NIV. The authors concluded that these results could be related to a reduction in the cost of breathing in patients supported with NIV, probably by reducing the ventilatory burden imposed on the inspiratory neck muscles to balance diaphragm weakness.

Similar results were found by Barle et al. [24], who aimed to elucidate the difference in the REE between SB and mechanical ventilation. The authors found that invasive bi-level pressure support ventilation (PSV) reduced the oxygen cost of breathing in long-standing tracheostomized post-polio patients compared with bi-level controlled mechanical ventilation (CMV) as well as during SB. Furthermore, the authors did not find any differences between the REE measured during SB and the REE measured during bi-level CMV. However, these results reflect the fact that a certain degree of hyperventilation may have prevailed to ensure passive respiration, due to the specific pathological condition of the patients recruited [24].

Compared to our study and to Sirala et al. [20], these two studies analyzed subjects with chronic conditions that require ventilatory support, particularly at the end-stage. In contrast, the patients enrolled in our study were children admitted to the PICU for acute respiratory disease and weaning from NIV, with no chronic illness.

Overall, all the studies analyzed agree on the potential of employing IC in the canopy mode to measure the REE in subjects undergoing NIV.

One limitation of our study was the small sample size. The reason for the low number of subjects enrolled was the slow recruitment due to the age of the subjects. Indeed, children under 6 years of age may have a lower tolerance of the canopy helmet compared to older children. However, it would have been difficult to enroll subjects older than 6 years of age, as the use of NIV for the treatment of acute respiratory illness in our PICU is generally limited to the treatment of young children under 6 years of age [25].

Another limitation of our study was that we were only able to test IC-guided REE measurement during NIV only at room air, i.e., with a fraction of inspired oxygen (FiO_2_) equal to 0.21. The reason behind this limitation is that, at the present time, the IC canopy mode is only validated to perform IC in subjects at room air. Therefore, in the canopy mode, it is not possible to measure REE in subjects undergoing NIV with a FiO_2_ > 0.21. On the other hand, the IC mechanical ventilation mode cannot be employed in subjects who undergo NIV because, in this setting, the IC gas analyzer is directly connected to the ventilator circuit, which is a closed circuit [1,3].

## 5. Conclusions

Using IC to measure the REE in children undergoing NIV could be feasible. Future prospective studies should define a standard method to optimize the performance of IC in patients undergoing single-limb NIV.

## Figures and Tables

**Figure 1 nutrients-16-00230-f001:**
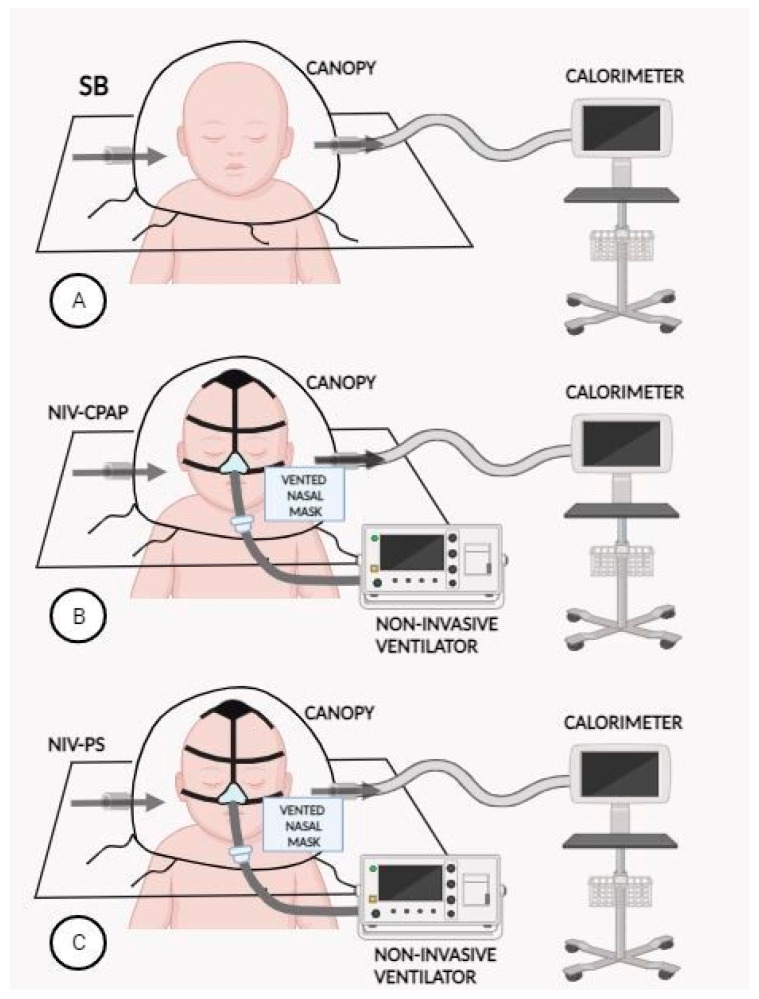
Indirect calorimetry performed with canopy mode. (**A**) Spontaneous breathing, (**B**) during NIV-CPAP, (**C**) during NIV-PS.

**Figure 2 nutrients-16-00230-f002:**
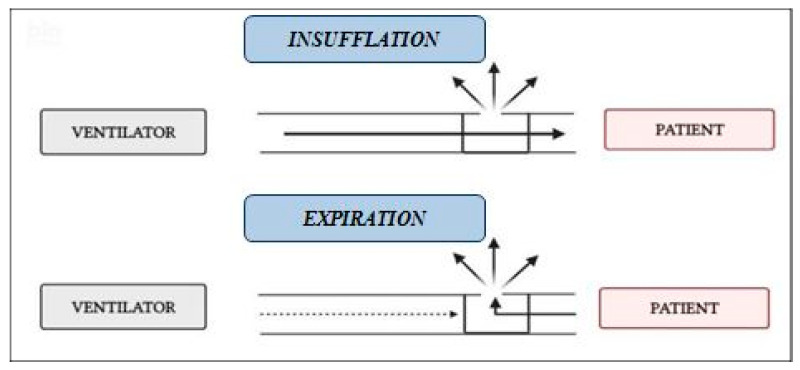
Single-limb with intentional leak circuit. Leaks may be positioned in the circuit (whisper) or in the mask (vented mask).

**Figure 3 nutrients-16-00230-f003:**
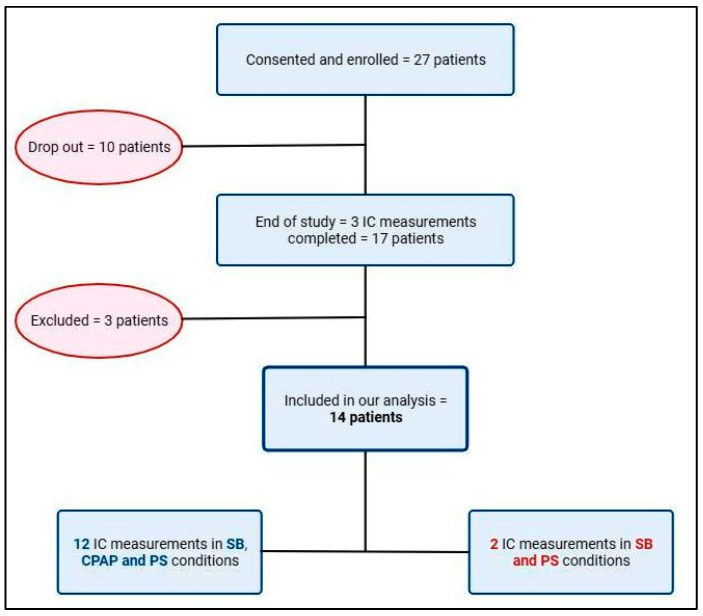
Study flow chart: patient’s enrollment. Abbreviations: IC = indirect calorimetry; SB = spontaneous breathing; CPAP = continuous positive airway pressure; PS = pressure support.

**Figure 4 nutrients-16-00230-f004:**
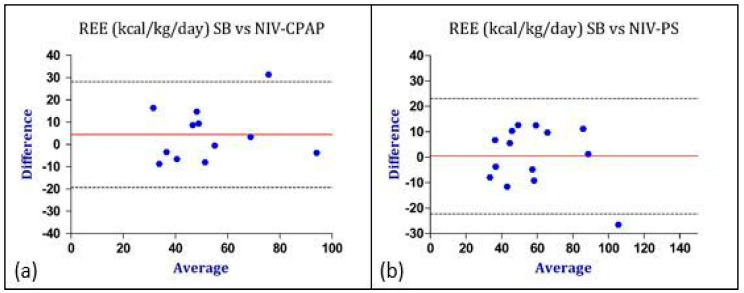
Graphical representation of Bland–Altman analysis. Legend: red line: bias; discontinuous lines: upper and lower LoA. (**a**) Comparison of REE between SB and NIV-CPAP; (**b**) comparison of REE between SB and NIV-PS. The “Y” axis represents the difference between the REE measured in SB vs. the REE measured during NIV-CPAP and NIV-PS. The “X” axis represents the average of the REE measured in SB and the REE measured during NIV-CPAP and NIV-PS. Data are expressed as absolute values (kcal/kg/day). Abbreviations: REE = resting energy expenditure; SB = spontaneous breathing; CPAP = continuous positive airway pressure; PS = pressure support.

**Figure 5 nutrients-16-00230-f005:**
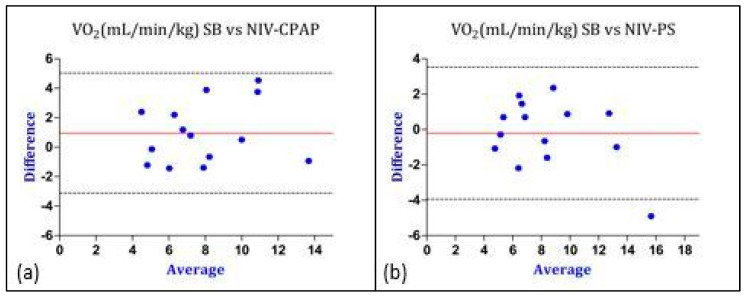
Graphical representation of Bland–Altman analysis. Legend: red line: bias; discontinuous lines: upper and lower LoA. (**a**) Comparison of VO_2_ between SB and NIV-CPAP; (**b**) comparison of VO_2_ between SB and NIV-PS. The “Y” axis represents the difference between the VO_2_ measured in SB vs. the VO_2_ measured during NIV-CPAP and NIV-PS. The “X” axis represents the average of the VO_2_ measured in SB and the VO_2_ measured during NIV-CPAP and NIV-PS. Data are expressed as absolute value (mL/min/kg). Abbreviations: VO_2_ = oxygen consumption; SB = spontaneous breathing; CPAP = continuous positive airway pressure; PS = pressure support.

**Figure 6 nutrients-16-00230-f006:**
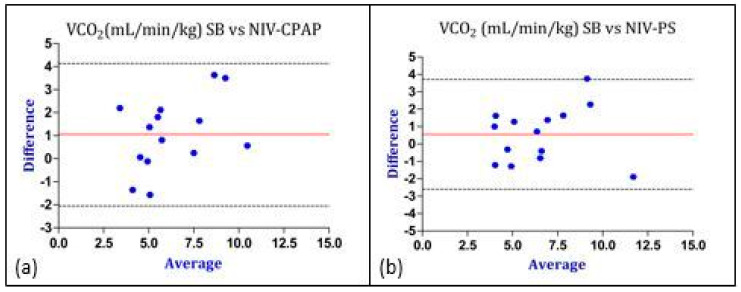
Graphical representation of Bland–Altman analysis. Legend: red line: bias; discontinuous lines: upper and lower LoA. (**a**) Comparison of VCO_2_ between SB and NIV-CPAP; (**b**) comparison of VCO_2_ between SB and NIV-PS. The “Y” axis represents the difference between the VCO_2_ measured in SB vs. the VCO_2_ measured during NIV-CPAP and NIV-PS. The “X” axis represents the average of the VCO_2_ measured in SB and the VCO_2_ measured during NIV-CPAP and NIV-PS. Data are expressed as absolute value (mL/min/kg). Abbreviations: VCO_2_ = carbon dioxide production; SB = spontaneous breathing; CPAP = continuous positive airway pressure; PS = pressure support.

**Figure 7 nutrients-16-00230-f007:**
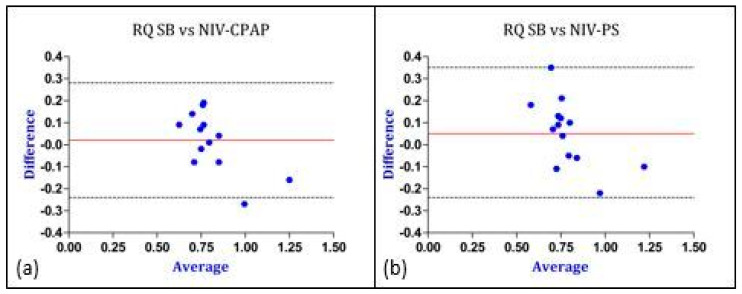
Graphical representation of Bland–Altman analysis. Legend: red line: bias; discontinuous lines: upper and lower LoA. (**a**) Comparison of RQ between SB and NIV-CPAP; (**b**) comparison of RQ between SB and NIV-PS. The “Y” axis represents the difference between the RQ measured in SB vs. the RQ measured during NIV-CPAP and NIV-PS. The “X” axis represents the average of the RQ measured in SB and the RQ measured during NIV-CPAP and NIV-PS. Data are expressed as absolute value (VCO_2_/VO_2_). Abbreviations: RQ = respiratory quotient; SB = spontaneous breathing; CPAP = continuous positive airway pressure; PS = pressure support.

**Table 1 nutrients-16-00230-t001:** Descriptive statistics of vital signs.

	SB			NIV-CPAP			NIV-PS		
	HR (bt/min)	RR (bts/min)	SpO_2_ (%)	HR (bt/min)	RR (bts/min)	SpO_2_ (%)	HR (bt/min)	RR (bts/min)	SpO_2_ (%)
*n*	13	13	13	13	13	12	14	14	14
1 QT	101.0	31.00	93.00	99.50	27.00	92.25	89.25	27.75	95.00
Median	140.0	40.00	93.00	130.0	36.00	94.00	118.0	36.00	95.00
3 QT	171.5	56.00	96.00	160.0	62.00	95.75	150.3	59.50	98.00
Mean	138.1	43.00	94.15	129.2	43.38	94.17	123.2	43.07	95.71
SD	35.50	15.89	2.911	33.18	18.48	3.040	34.05	22.39	3.173

Abbreviations: HR = heart rate; RR = respiratory rate; SpO_2_ = oxygen saturation; SB = spontaneous breathing; NIV = non-invasive ventilation; CPAP = continuous positive airway pressure; PS = pressure support.

**Table 2 nutrients-16-00230-t002:** Paired *t*-Test of vital signs and COMFORT-B measurement.

SB vs. CPAP	*p*-Value	SB vs. PS	*p*-Value
HR (bt/min)	0.02	HR (bts/min)	0.004
RR (bts/min)	1.0	RR (bts/min)	0.432
SpO_2_ (%)	0.90	SpO_2_ (%)	0.2445
COMFORT-B	0.82	COMFORT-B	0.24

Abbreviations: HR = heart rate; RR = respiratory rate; SpO_2_ = oxygen saturation; SB = spontaneous breathing; CPAP = continuous positive airway pressure; PS = pressure support.

**Table 3 nutrients-16-00230-t003:** Primary endpoint: comparison between SB and NIV-CPAP.

	SB *	CPAP *	Paired *t*-Test	Bland–Altman Analysis		
*n*	12	12	*p*-Value	BIAS	Lower LoA	Upper LoA
VCO_2_ [mL/min/kg]	6.8 ± 2.5	5.8 ± 1.9	0.03	1.06	−2.02	+4.14
VO_2_ [mL/min/kg]	8.4 ± 3.1	7.4 ± 2.7	0.11	0.96	−3.12	+5.04
RQ	0.82 ± 0.1	0.80 ± 0.2	0.6	0.02	−0.24	+0.28
REE [kcal/kg/day]	58.04 ± 21.1	50.4 ± 18.5	0.23	4.41	−19.4	+28.23

Abbreviations: SB = spontaneous breathing; CPAP = continuous positive airway pressure; LoA = Limit of Agreement. * are expressed as mean ± standard deviation.

**Table 4 nutrients-16-00230-t004:** Secondary endpoint: comparison between SB and NIV-PS.

	SB *	PS *	Paired *t*-Test	Bland–Altman Analysis		
*n*	14	14	*p*-Value	BIAS	Lower LoA	Upper LoA
VCO_2_ [mL/min/kg]	6.8 ± 2.5	6.2 ± 2.3	0.2	0.55	−2.56	+3.67
VO_2_ [mL/min/kg]	8.4 ± 3.1	8.6 ± 3.8	0.7	−0.2	−3.94	+3.54
RQ	0.82 ± 0.1	0.76 ± 0.2	0.2	0.05	−0.24	+0.35
REE [kcal/kg/day]	58.04 ± 21.1	57.7 ± 23.9	0.9	0.4	−22.23	+23.07

Abbreviations: SB = spontaneous breathing; PS = pressure support; LoA = Limit of Agreement. * are expressed as mean ± standard deviation.

## Data Availability

The data presented in this study are available on request from the corresponding author. The data are not publicly available due to privacy reasons.

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
