# Peer review of "Validation of Indirect Calorimetry in Children Undergoing Single-Limb Non-Invasive Ventilation: A Proof of Concept, Cross-Over Study"

_nutrients, 2024, doi:10.3390/nu16020230_

Round 1

Reviewer 1 Report

Comments and Suggestions for Authors

Review comments for Nutrients-2761169

Title: Validation of Indirect Calorimetry in children undergoing single-limb non-invasive ventilation: a proof of concept, cross-over study

Summary
The authors have conducted a cross-over study to investigate the feasibility of indirect calorimetry in critically ill children requiring non-invasive mechanical breathing. 

General Comments:

·         Formatting of the article should be revised extensively. 

·      Basic knowledge of indirect calorimetry and its results should be acknowledged; many misleading statements as listed in “specific comments” below.

·         Perhaps a comparison with an established equation for calculating BMR or REE may help demonstrate the feasibility of the measurement of SB condition, as canopy measurements itself is challenging in this population.

·         10/24 citations are from > 10 years before; suggest updating refences.

·         English proofreading is advised.

Specific comments:

Line 53: REE described here represents basic metabolic rate (BMR) as described in reference #3. 

Line 56: “indirect calorimetry” is not the “direct measurement” of REE, but rather an indirect measurement of heat production through the measurement of oxygen consumption and carbon dioxide production. I suggest removing the word “direct” as it seems misleading.

Line 83-110: Explanation of recently published studies in the “Introduction” seems redundant and should be provided in “Discussions” instead.

Figure 1: The figure illustrates standard canopy mode IC testing; the setting for NIV measurements would be more attractive to be seen as a figure. Consider replacing or adding another illustration of NIV mode measurements.

Lines 151-152, 154-156: Explanation of supposed effects should not be provided in the Methods section; either move to the “Introduction” as background knowledge or more preferably to “Discussions” as a part of the interpretation of the result.

Lines 182-183: A single-arm study does not involve randomization. Please omit.

Line 253: “gold standard” should be “SB” to match the title with Table 3

Line 344: CPAP mode without controlled ventilation is not likely to cause opposition if not discomfort.

Line 354: “By contrast” should read “In contrast”

Lines 360-364: Redundant punctuation (commas); please revise

Lines 390-398: Conclusion should be stated on the results of the current study. Most of the elements written here belong to the Discussions section; please revise.

Comments on the Quality of English Language

Redundant writing and punctuation errors are observed especially (but not limited) in the Dicussions section.

Reviewer 2 Report

Comments and Suggestions for Authors

The study “Validation of Indirect Calorimetry in children undergoing single-limb non-invasive ventilation: a proof of concept, cross-over study” by Veronica D’Oria et al. examined IC during spontaneous breathing (SB), NIV-CPAP and NIV-PS in PICU patients. The study aimed to validate the measurement of IC in pediatric patients undergoing non-invasive ventilation through a single-limb circuit with intentional leaks. The main endpoints were to verify the agreement of REE during spontaneous breathing (gold standard) and NIV-CPAP or NIV-PS mode. Bland-Altman analysis was employed to compare REE, V̇O2, V̇CO2, and respiratory quotient (RQ). The authors concluded that the findings support the accuracy of IC in children undergoing NIV. The study is a novel one and of importance. There are, however, major limitations and weaknesses that should be corrected.

Abstract

·       A brief description of the method of IC measuring is missing.

Introduction

L48 Recent insights have underscored the adverse impact of malnutrition, encompassing both underfeeding and overfeeding, on patient outcomes.

·       Please insert a relevant paediatric reference. 

52 Thereby emphasizing the critical importance of assessing resting energy expenditure (REE) in PICU-admitted patients [3-9]. 

·       Although the sentence refers to PICU-admitted patients, references 3-9 are articles related to adult patients and most of them are very old (2011 – 2015). Less relevant or very old should be replaced by the recent studies related to the importance of accurately assessing REE in PICU patients.

L54 However, several studies have demonstrated that commonly used predictive equations often provide inaccurate estimates, leading to under- or overestimation of energy requirements [10].

·       Although this is an old but very good reference, it is one not several. Please add the following reference:

    Nutrients. 2022 Oct 10;14(19):4211. doi: 10.3390/nu14194211.

L84 A recent study, aimed at assessing the accuracy of indirect calorimetry (IC) in individuals supported by NIV with an intentional leaks circuit, was recently published [19].

·       2012 is not recent. Please delete the word “recent”. 

Methods

L142 NIV was carried out by a single-limb circuits with intentional leaks using a turbine-driven ventilator.

·       The paper would benefit from a figure of a schematic presentation of the method used in patients on NIMV including circuit, canopy, ventilator, and mask. 

L142 We chose the vented nasal mask…

·       Why not an oral-nasal or the commonly used full-face mask? Please explain.

L207 The reasons of PICU admission were Bronchiolitis (6 subjects), Pneumonia (4), Post surgery respiratory distress (3), Laryngotracheitis (1).

·       NIV IC is of importance in chronically ill patients, whose nutrition needs are difficult to estimate, but not in acutely ill patients receiving no supplementary oxygen, who will start oral feeds the next day. Nobody would measure IC in these patients. It is not understandable why the authors included patients with these common diseases and not patients with neuromuscular or other chronically ill patients.

Discussion

L281 Although there was no significant difference between the calculated COMFORT-B score for the two ventilatory mode compared to SB (p>0.05), considering the limit of the evaluation scores in children.

·       Is something missing? Otherwise, delete “Although”.

L317 Limitations

·       Studying patients with acute rather than chronic conditions is a clinically relevant limitation. Also, the use of nasal rather than full-face masks is another one since most patients are now supported with full-face masks. Finally, the study did not assess IC in different clinical settings (FiO2>21%, different rates, or pressures). These limitations should be acknowledged and discussed. 

Conclusions

·       This is a discussion, and the sentences should be moved to relevant paragraphs in the discussion. The conclusion should only conclude with the results of the study. Such a conclusion is in the abstract. Remember, the limitations of this study do not permit extrapolating conclusions in clinical practice. 

Comments on the Quality of English Language

Minor English editing is needed.

Round 2

Reviewer 1 Report

Comments and Suggestions for Authors

General Comments

The authors have made significant improvements to the manuscript according to the reviewer comments. Only minor editing issues to be noted.

Specific Comments

Lines 97-110: Paragraphs should be re-organized. Lines97-101, lines 102-110 should be a single paragraph, lines 104-108 can be moved to “Methods” section under “2.5 Data analysis” (line 183)

Line 353 and 359: “limit” should read “limitation”

Comments on the Quality of English Language

Written in comments

Reviewer 2 Report

Comments and Suggestions for Authors

Figure 3 is of low quality. In figures 4-7, the labels should indicate the units i.e. how the difference or average is calculated (i.e. divided by a parameter or 100 or whatever) and is it % or absolute values? Finally, most review references should be replaced by recent original studies in PICU patients (like the ones you have already added) to increase the scientific background of the article. When there is recent original research in the field it should be preferred instead of reviews (usually based on older studies).    

Comments on the Quality of English Language

A few minor errors
